# Triple Negative Breast Cancer Preclinical Therapeutic Management by a Cationic Ruthenium-Based Nucleolipid Nanosystem

**DOI:** 10.3390/ijms24076473

**Published:** 2023-03-30

**Authors:** Maria Grazia Ferraro, Marco Bocchetti, Claudia Riccardi, Marco Trifuoggi, Luigi Paduano, Daniela Montesarchio, Gabriella Misso, Rita Santamaria, Marialuisa Piccolo, Carlo Irace

**Affiliations:** 1BioChemLab, Department of Pharmacy, School of Medicine and Surgery, University of Naples “Federico II”, Via D. Montesano 49, 80131 Napoli, Italy; mariagrazia.ferraro@unina.it (M.G.F.); carlo.irace@unina.it (C.I.); 2Biogem Scarl, Institute of Genetic Research, Laboratory of Molecular and Precision Oncology, 83031 Ariano Irpino, Italy; marco.bocchetti@unicampania.it; 3Department of Precision Medicine, School of Medicine and Surgery, University of Campania “Luigi Vanvitelli”, 80138 Naples, Italy; gabriella.misso@unicampania.it; 4Department of Chemical Sciences, University of Naples “Federico II”, Via Cintia 21, 80126 Napoli, Italy; claudia.riccardi@unina.it (C.R.); marco.trifuoggi@unina.it (M.T.); luigi.paduano@unina.it (L.P.);

**Keywords:** ruthenium(III) complex, nucleolipid nanosystem, DOTAP liposome, triple-negative breast cancer (TNBC), preclinical investigations, anticancer activity, cell migration and invasion

## Abstract

Based on compelling preclinical evidence concerning the progress of our novel ruthenium-based metallotherapeutics, we are focusing research efforts on challenging indications for the treatment of invasive neoplasms such as the triple-negative breast cancer (TNBC). This malignancy mainly afflicts younger women, who are black, or who have a BRCA1 mutation. Because of faster growing and spreading, TNBC differs from other invasive breast cancers having fewer treatment options and worse prognosis, where existing therapies are mostly ineffective, resulting in a large unmet biomedical need. In this context, we benefited from an experimental model of TNBC both in vitro and in vivo to explore the effects of a biocompatible cationic liposomal nanoformulation, named HoThyRu/DOTAP, able to effectively deliver the antiproliferative ruthenium(III) complex AziRu, thus resulting in a prospective candidate drug. As part of the multitargeting mechanisms featuring metal-based therapeutics other than platinum-containing agents, we herein validate the potential of HoThyRu/DOTAP liposomes to act as a multimodal anticancer agent through inhibition of TNBC cell growth and proliferation, as well as migration and invasion. The here-obtained preclinical findings suggest a potential targeting of the complex pathways network controlling invasive and migratory cancer phenotypes. Overall, in the field of alternative chemotherapy to platinum-based drugs, these outcomes suggest prospective brand-new settings for the nanostructured AziRu complex to get promising goals for the treatment of metastatic TNBC.

## 1. Introduction

Anticancer Ru-based metallochemotherapeutics rightfully hold a foreground place among the most studied alternative to non-platinum derivatives. Indeed, in the last decades, a huge number of antiproliferative Ru-based complexes have been subjected to in-depth preclinical studies, and the literature reveals that progress in perspective Ru-based anticancer agents is outstanding (for recent reviews, see refs. [1,2,3,4]). Primarily, both organic and inorganic Ru(II) and Ru(III) complexes have been, so far, proposed, sharing special electronic and steric features within a wide repertoire of possible chemical ligands and decorations. Ru(II/III) complexes are normally hexacoordinated, providing an octahedral configuration, and are thermodynamically and kinetically stable in physiological conditions, allowing for straightforward applications in medicinal chemistry [5]. Ru(II) derivatives are typically more reactive and bioactive and, accordingly, can engender a range of cellular effects, whereas Ru(III) complexes offer the advantage of being activated in situ within the tumor biological microenvironment, thereby performing bioactivity in a more selective fashion (a process known as activation by reduction) [3,5,6]. While belonging to the group of so-called platinoids, the variety of Ru-containing molecular platforms originated by exploiting diverse eligible ligands can promote, in turn, the interaction with a number of biological targets, supporting the possibility of multiple modes of action different from cisplatin and congeners [7]. Accordingly, with regard to the field of anticancer research, different molecular mechanisms, both cellular and extracellular, have been so far assumed as the basis of the action of many ruthenium derivatives, now considered multitarget-acting compounds [2,8].

Aiming at the development of non-canonical metal-based anticancer drugs, our research team has profitably blended the physico-chemical properties of biocompatible nucleolipid-containing nanomaterials with the pharmacological ones provided by original ruthenium(III) complexes endowed with superior antiproliferative activities. Moving in this direction, we developed different classes of Ru(III) nucleolipid complexes, largely described in the literature [8,9,10]. Their co-aggregation with lipids featured by different biological properties finally produced stable liposomal nanosystems (see Appendix A for additional structural details), which proved to be particularly effective in anticancer preclinical evaluations [11,12,13,14]. Within the suite of established derivatives, the positively charged nanosystems (based on the cationic lipid DOTAP) have, hitherto, showcased the most favorable performance in terms of both selective and effective inhibition of tumor growth [15,16]. Meanwhile, a very promising outcome has been achieved in the control of the growth and proliferation of breast cancer cells (BCC), where both nuclear and cytosolic targeting seemed capable of activating programmed cell death (PCD) pathways by a change in the expression profile of regulatory proteins, e.g., bcl-2 protein family [8,17]. The so-called HoThyRu/DOTAP formulation (see Appendix A for molecular structures)—one of our most active nanosystems containing the AziRu ruthenium(III) complex as the starting scaffold—demonstrated safety throughout preclinical investigations in animal models, as well as the ability to restore cell death pathways, i.e., apoptosis and pro-death autophagy, normally suppressed in several human tumor phenotypes [8,18]. In order to deepen knowledge concerning the biological behavior of our candidate drugs, as well as to further unveil their action, efficacy, and safety profile, here we investigate the biological effects of the cationic HoThyRu/DOTAP nanosystem exploiting preclinical models of triple-negative breast cancer (TNBC) [19]. Indeed, for this study, we have benefited from using an in vitro cellular model (MDA-MB-231 cells) on behalf of a definite TNBC phenotype to generate an in vivo xenograft model of human BCC [20]. TNBC is an invasive carcinoma and aggressive metastatic disease with a poor prognosis and fewer available treatment options than other BC subtypes. The malignant nature of this disease stems from its unique dissemination pattern. It is commonly typified as estrogen receptor-negative, progesterone receptor-negative, and human epidermal growth factor receptor 2-negative, accounting for about 10–20% of all BC and mainly affecting with higher frequency younger premenopausal and/or black women, as well as patients with BRCA1 and BRCA2 germline mutations (BRCA altered status) [21,22]. Chemotherapy still remains the mainstay of TNBC management, especially in case of metastatic spread, but it is nowadays significantly limited to the use of a few drugs (including some platinum derivatives) due to low responsiveness and chemoresistance of tumor phenotypes, as well as systemic toxicity and low bioavailability [23,24]. While emerging pathways and therapeutic strategies are beneath investigation to expand TNBC clinical management, including immunotherapies and poly(ADP-ribose) polymerase (PARP) inhibitors [25], disposal of new effective unconventional metallochemotherapeutics, possibly free from significant side effects, would be a major research goal in the perspective of novel anticancer curative options for TNBC.

## 2. Results

### 2.1. Antiproliferative Effect In Vitro

We first analyzed the antiproliferative effect of HoThyRu/DOTAP nanoformulation on the MDA-MB-231 triple-negative phenotype by determination of the “cell survival index” through targeted bioscreens. Incubations were performed considering the actual ruthenium concentrations in the HoThyRu/DOTAP nanosystem (effective metal concentration of 30% mol/mol). Data reported in Figure 1a show a significant reduction in cell survival after 48 h of exposure to HoThyRu/DOTAP. The IC_50_ value in MDA-MB-231 cells, referred to the actual AziRu concentration loaded in the HoThyRu/DOTAP nanosystem, is in the low micromolar range (around 9 µM) and is roughly similar to those of cisplatin (around 7 µM), here used as a reference anticancer drug. As expected, under the same experimental conditions, the bare AziRu complex does not produce a biological effect on cell survival. Calculated IC_50_ values for the total liposomal nanoformulation (HoThyRu/DOTAP), the loaded ruthenium(III) complex (AziRu), the naked AziRu, and cisplatin (*c*DDP) are reported in Figure 1b. Biological effects examined in healthy cells as control cultures, i.e., keratinocytes (HHFKs) and fibroblasts (HDFa), were significantly attenuated with respect to cisplatin used in the same experimental conditions as the reference drug.

### 2.2. Clonogenic Assay

The biological activity of HoThyRu/DOTAP was validated via the colony formation assay in the experimental model of TNBC. As shown in Figure 2, a significant reduction in the ability of single cells to survive and reproduce, forming colonies, was detected after 48 h of incubation at the IC_50_ concentration with respect to untreated cultures. Again, the capacity of HoThyRu/DOTAP to interfere with the growth and proliferation of MDA-MB-231 cells was similar to cisplatin.

### 2.3. Ruthenium Cellular Uptake and Intracellular Biodistribution in MDA-MB-231 Cells

Next, cellular uptake of HoThyRu/DOTAP at the IC_50_ concentration (30 µM, i.e., 9 µM of AziRu) was evaluated following in vitro incubation for 24 h. The analysis via ICP-MS performed as specified in the experimental section revealed an almost quantitative cellular uptake of HoThyRu/DOTAP. In fact, over 80% of the total ruthenium loaded in the nanoformulation and used for incubations was detected in MDA-MB-231 cells. Concerning the intracellular ruthenium pool after cellular uptake, about 63% and 37% were found in the cytosol and in nuclei, respectively, as showcased in Figure 3. In turn, very significant fractions of intracellular ruthenium have been revealed in mitochondria (62% of ruthenium amount found in cytosolic fraction) and nuclear DNA (53% of ruthenium amount found in nuclear fraction), indicative of a large and homogeneous distribution within the cells.

### 2.4. Cell Death Pathways Activation

Fluorescence studies, performed by means of specific fluorescent probes and confocal microscopy, as thoroughly described in the experimental section, revealed significant activation of PCD pathways after exposure of MDA-MB-231 cells to the IC_50_ concentration of HoThyRu/DOTAP for 48 h. As underscored in fluorescence microphotographs (Figure 4a), treatment of cells with HoThyRu/DOTAP caused a marked activation of apoptosis. The probe associated with the green fluorescence signal selectively recognizes the membrane PS, whose exposure to the outer plasma membrane is a feature of apoptosis, enabling recognition and phagocytosis. The analysis of the percentage of positive green fluorescence cells shows results rather similar to those obtained after cisplatin application in vitro (Figure 4b). Conversely, no evidence of necrosis (red fluorescence) was detectable. In the same experimental model, fluorescence analysis for the evaluation of autophagy activation (green fluorescence signal) highlighted a marked appearance of autophagic vacuoles (i.e., autophagosomes and autophagolysosomes), especially in the perinuclear cytosolic regions of treated MDA-MB-231 cells (Figure 5a,b). Rapamycin was used as the positive control for autophagy activation. 

### 2.5. Antimetastatic Effect In Vitro

The effects of HoThyRu/DOTAP nanoformulation on the invasion and migration ability of MDA-MB-231 cells were subsequently explored. Before performing the functional assays, we selected by preliminary experiments a sub-IC_50_ concentration (24 μM of HoThyRu/DOTAP in vitro, corresponding to 7.2 µM of AziRu) to set appropriate experimental conditions, being sustained cell death, a factor critically interfering with cell invasion and migration. Moreover, since cell migration can be influenced by culture medium, pilot tests were performed by progressively decreasing serum concentration (serum starvation) to optimize experimental conditions. However, to avoid interference and consider that serum deprivation maximizes MDA-MB-231 invasion, cell migration experiments were performed under serum starvation conditions [26]. The Abcam Cell Invasion Assay (Collagen I) 24-well plate was used to assess both invasions with a membrane coated with a Collagen I matrix and migration with an uncoated membrane. As shown in Figure 6, a significant downregulation of cellular invasion and migration was observed after 48 h of HoThyRu/DOTAP treatment at the selected sub-IC_50_ concentration. Following this path, collective cell migration ability was determined by wound healing assay. As presented in Figure 7a, the migration capacity of MDA-MB-231 cells, monitored by phase contrast microscopy, was significantly reduced in the presence of a sub-IC_50_ concentration of HoThyRu/DOTAP. Accordingly, the line graph in Figure 7b, which plots the percentage of wound closure, proves a considerably reduced migratory capacity of the cells starting at 48 h from the beginning of the experiment, with an overall reduction in migration ability of about 50% at the endpoint (96 h). Hence, these data clearly indicate that the HoThyRu/DOTAP nanoformulation is able to inhibit migratory cells affecting invasion processes and wound field closure.

### 2.6. Analysis of a Limited Panel of EMT Markers by RT-qPCR

To give an insight into the antimetastatic effect of HoThyRu/DOTAP, the expression pattern of recognized markers for epithelial-mesenchymal transition (EMT), typically associated with several tumorigenic events, including metastasis, was studied. E- and N-cadherins, vimentin, Slug, and Snail were analyzed by means of real-time quantitative polymerase chain reaction (qPCR), investigating whether the mRNA expression levels were affected by treatments in vitro. As represented in Figure 8, the E-cadherin gene has an increased expression after 48 h of treatment with HoThyRu/DOTAP sub-IC_50_ concentration, while the N-cadherin gene has no significant variations compared to untreated cells (control). Considering that the up-regulation of N-cadherin followed by the down-regulation of E-cadherin is a conventional hallmark of EMT, this outcome supports a molecular interference by the ruthenium complex with EMT throughout tumorigenic processes such as cell migration and invasion. Under the same experimental conditions, we did not observe significant variations of vimentin, a type III intermediate filament involved in cell adhesion, motility, and migration. Finally, based on these results, the mRNAs expression of the transcriptional repressors Slug and Snail was evaluated. A substantial down-regulation after HoThyRu/DOTAP application to cells was detected, suggestive of E-cadherin regulation by these repressors. These findings are well-matching with the overall decrease in MDA-MB-231 cells’ migration ability induced by the HoThyRu/DOTAP treatment.

### 2.7. Anticancer Effect In Vivo

To explore HoThyRu/DOTAP anticancer efficacy in a TNBC model in vivo, xenografts were established by s.c. injection of MDA-MB-231 cells into the right flank of nude mice. Two weeks post-tumor implant, and after accurate verification of tumor development, animals were enrolled for the in vivo study. As reported in Figure 9a, the experimental protocol has been envisioned for i.p. administration of HoThyRu/DOTAP at the dose of 15 mg/kg once a week for 28 days. The endpoint of the study was at five weeks starting from the first treatment. As clearly shown in Figure 9d,e, HoThyRu/DOTAP formulation was able to considerably reduce the weight and volume of the tumor masses in all treated animals to the indicated dosage regimen. These outcomes are supported by photographic evidence of animals during the in vivo trial (see Figure 9f,g). Conversely, mice to which the HoThyRu/DOTAP formulation was not administered (untreated xenotransplanted) developed clearly evident large subcutaneous tumors (Figure 9g). In the same animal experimental model, we have already checked the biological effects of the administrations of the pure DOTAP liposome and the not co-aggregated HoThyRu nucleolipid complex, which were not associated with a detectable impact on tumors [18]. In regards to the safety profile, animal survival was 100% for the mice group treated with HoThyRu/DOTAP (Figure 9b). Moreover, no alteration in body weights was recorded (Figure 9c), as well as no signs of animal suffering were observed, suggesting the therapeutic schedule was well tolerated in vivo. 

### 2.8. Ruthenium Bioaccumulation in Mice and Xenograft Tumor Lesions 

We finally analyzed ruthenium accumulation in tumour samples appropriately collected at the end-point of the in vivo study after systemic administration (i.p.) of HoThyRu/DOTAP. Samples deriving from the tumor lesions were subjected to a specific procedure and subsequently investigated by ICP-MS, as described in the experimental section. In compliance with our previous findings, data show significant ruthenium amounts in tumor samples, corresponding to about 15% of all the administered ruthenium via HoThyRu/DOTAP nanoformulation (Figure 10). Remarkably, the ruthenium bioaccumulation profile in vivo we have found in mice biological samples (heart, lung, spleen, kidney, and liver) following HoThyRu/DOTAP administrations is very similar to previously published data [18]. This evidence further supports the in vivo behavior of the HoThyRu/DOTAP nanoformulation. 

## 3. Discussion

In continuity with our investigations focused on the development of new Ru(III)-based anticancer drugs, the aim of this study was to explore at a preclinical level the ability of the cationic HoThyRu/DOTAP nanoformulation to inhibit the growth and proliferation of a human invasive TNBC phenotype. TNBCs represent roughly 15% of all breast cancer phenotypes and are associated with decreased overall survival, being the most aggressive and dangerous BCs to human health, enough to be regarded as one of the unmanageable types of BC [19,22]. From the standpoint of a very heterogeneous disease, TNBC subtypes lack expression of both estrogen (ER) and progesterone (PR) hormone receptors, as well as epidermal growth factor receptor 2 (HER2), thereby becoming much more challenging to understand and treat [21]. Moreover, the diagnosis usually comes when considerable dissemination of highly invasive metastatic cells has already occurred [27]. Accordingly, hormone therapy and drugs targeting HER2 are not supportive, so that chemotherapy remains one of the main systemic clinical options [23,25]. Although some progress has however been achieved in therapy, the overall decrease in the TNBC death rate observed in the last decade (especially in older women) is believed to be the result of early diagnosis through screening and increased awareness rather than better treatments. Indeed, while systemic chemotherapy is considered the mainstay of treatment options for patients with metastatic TNBC, its therapeutic effectiveness is mostly transitory, providing a reasonable explanation for a median survival that occasionally goes beyond 12–18 months. Moreover, conventional chemotherapy also suffers from additional limitations, e.g., toxicity, bioavailability, and resistance [28,29].

In the landscape of metal-based chemotherapy, many cisplatin congeners alone or in combination have been investigated in preclinical and clinical trials for TNBC management. The majority of patients with metastatic TNBC had, in fact, received platinum-containing chemotherapy, especially in the first-line treatment, frequently combined with other regimens, i.e., taxanes, anthracyclines, gemcitabine, 5-fluorouracil (5-FU), cyclophosphamide) [24,30,31]. An alternative route of research is represented by targeted therapies where various options are emerging in recent years, including poly-ADP-ribose polymerase (PARP), immune check-point and mTOR inhibitors, as well as antiangiogenic and immunotherapeutic agents [23,25,28,29]. However, despite intensive research, therapeutic options in TNBC are still limited. Thus, the need for a substantial advance in TNBC treatment is right now imperative. Moving in this direction, ruthenium-based complexes deserve special attention for their inherent features in the midst of unconventional metallodrugs. Besides being less toxic than platinum-containing agents, anticancer ruthenium complexes can act by a “multi-targeted” approach, ensuring both enhanced antitumor efficacy and reduced chemoresistance [2,6,7,8]. Management of heterogeneous TNBC phenotypes could benefit from effective curative agents acting simultaneously on multiple targets. This approach could thereby represent one of the most promising attempts of the forthcoming systemic chemotherapy [32,33].

As uncovered by our data, the ruthenium(III) complex AziRu—when stably and safely lodged in the cationic HoThyRu/DOTAP nanosystem—was found to be particularly bioactive in preclinical models of TNBC based on MDA-MB-231 cells. Its inhibitory effect on tumor growth and proliferation is comparable in vitro to that of cisplatin; notwithstanding, the biological effects observed in healthy cells are very mild compared to cisplatin. This outcome is in accordance with our previous findings suggestive of HoThyRu/DOTAP biocompatibility on different control cells, including the non-tumorigenic breast epithelial cell line MCF-10A [8,16,17]. Indeed, the AziRu complex behaves similar to a double pro-drug, undergoing hydrolysis via ligand exchange and subsequent reduction to ruthenium(II). This behavior can promote the metal center activation in a more selective way in cancer cells [2,8]. Moreover, biocompatibility concern gets further support in the absence of adverse effects throughout in vivo trials. Indeed, we have not observed so far toxicological responses on animals following HoThyRu/DOTAP in vivo administration, neither macroscopic nor by autopsy procedures or blood diagnostics, and mice have never shown symptoms of stress and/or pain under drug regimen [18]. As we have uncovered in former investigations, on account of its physico-chemical properties, the ruthenium content after HoThyRu/DOTAP intraperitoneal administration is widely distributed in the body, reaching by far the tumor sites where it can significantly accumulate [18]. This feature has been now validated by the evidence of significant ruthenium amounts in TNBC tumor masses following in vivo treatments. The positive nanosystem surface charge can be engaged in passive targeting effects toward the tumor microenvironment and cancer cells [34,35]. For their part, MDA-MB-231 cells are endowed with an extensive negative charge on their surface, enabling possible interactions with cationic liposomes [36,37]. With reference to the mechanisms of action underlining the anticancer effect, confocal fluorescence imaging proved that the candidate drug AziRu was able to inhibit tumor progression through a multimodal mechanism. The mode of action is conceivably based on interactions with both nuclear and cytosolic cellular biomolecular targets, confirming the activation of PCD pathways in response to HoThyRu/DOTAP treatments. Similarly, to other cell models of BC we have already examined, apoptosis reactivation was extensive and represents the main cause of MDA-MB-231 death in vitro [17]. Meanwhile, the simultaneous autophagy occurrence can be regarded as within the multitargeting action of this candidate drug [8,9]. On this topic, we have already debated about the activation of sustained autophagy as a possible further target for upcoming anticancer therapies [8,17,38,39]. Regarding the assumption of potential mitochondrial targets, previous studies on the Bcl-2 family protein expression profile suggest a modulation in PCD pathways towards a proapoptotic effect following in vitro treatments of BCC, including MDA-MB-231 cells [8,17]. In this study, we have also revealed that properly nanostructured, AziRu could also interfere with distinctive hallmarks of aggressive tumor phenotypes, i.e., migration and invasiveness. In line, the administration of HoThyRu/DOTAP significantly reduces the migration capacity of MDA-MB-231 cells, the most widely used TNBC cell line in metastatic breast cancer research [40]. This effect could be linked to the multi-targeting capacity of this candidate drug, probably through inhibition of intracellular pathways directly implicated in tumor phenotypes genesis. In this frame, new investigations are underway to explore whether and how this outcome is associated with drug-dependent direct cytotoxicity or with a specific intracellular impact on druggable targets. The here-obtained results showed that the AziRu-dependent effect on MDA-MB-231 cells migration and mobility observed at sub-IC_50_ concentrations could be indicative of interferences with the pathways orchestrating the epithelial-to-mesenchymal transition (EMT), which proved to be critical in promoting invasive and migratory phenotypes [41]. In compliance, a preliminary analysis of some migration-related genes uncovered an interesting E-cadherin upregulation in response to HoThyRu/DOTAP application in vitro, whereas the E-cadherin to N-cadherin switch is assumed as a crucial hallmark of EMT within a complex signaling pathways network [42,43]. As further confirmation, we found a parallel downregulation of the transcriptional factors Snail and Slug, deemed at the core of signaling regulations of EMT and acting as direct repressors of E-cadherin [44,45]. Due to increasing evidence making EMT a crucial player in tumorigenesis, its prospective targeting is by now considered therapeutic interest in cancer [41]. For sure, more accurate studies will be needed to understand the possible role of AziRu in this process. Interestingly, such a type of bioactivity has already been reported for other investigated ruthenium complexes. The well-known NAMI-A, which has inspired the design of many ruthenium(III) complexes, including AziRu itself, has been considered for a long time as a selective anti-metastatic agent for its ability to significantly affect tumor-derived metastatic cells [46,47]. Indeed, it has been demonstrated that in vivo NAMI-A does not inhibit primary tumor growth while significantly decreasing the dissemination rate during metastasis formation [48]. However, this effect seems to be mainly related to an extracellular action, being NAMI-A able to interact with collagen and inhibit the matrix metalloproteinases MMP-2 and MMP-9 without disclosing a significant ability to enter cells [49]. In compliance, other Ru-based derivatives, such as some arene ruthenium(II) complexes, have more recently demonstrated to be active directly on MDA-MB-231 triple-negative cells, exerting a suppressive action on metastases via AKT signal pathway inhibition and significantly reducing cell migration, invasiveness, and adhesion, as well as angiogenesis [50,51,52]. In a broad overview, these pieces of evidence endorse Ru-based complexes to act as multimodal functional agents by inhibiting tumor proliferation as well as migration, invasion, and metastasis formation [7,8,34,40,53].

Despite the amount of available data, to date, only a few Ru-based therapeutics—i.e., NAMI-A, KP1019, and its derivative KP1339/BOLD-100—have advanced in clinical development compared to the number of the investigational derivatives and their potential biomedical applications (see Appendix A for molecular structures). Based on compelling preclinical evidence, we can realistically assume that the HoThyRu/DOTAP biocompatible nanosystem is a future candidate drug for clinical trials [54,55,56,57]. Moving in this direction, prospective new scenarios could open for the nanostructured AziRu to get to new curative options in the field of alternative chemotherapy to platinum-based drugs. In the meantime, upcoming developments related to the nanosystem design and decorations for active targeting toward human BCC could further improve efficacy and safety [58]. Further investigations are ongoing to uncover the molecular mechanisms of action of AziRu, while in-depth SAR studies will try to shed light on its biomolecular targets and interactions. 

## 4. Materials and Methods

### 4.1. HoThyRu/DOTAP Liposome Preparation

The ruthenium(III) complex, named HoThyRu, was prepared by reacting in stoichiometric amounts the starting nucleolipid HoThy with the Ru complex Na^+^ [trans-RuCl_4_(DMSO)_2_]^−^ following a previously described procedure [13,15]. The final nucleolipidic Ru(III) complex was obtained in a pure form, as confirmed by TLC, and almost quantitative yields. The obtained spectroscopic (^1^H- and ^13^C-NMR) and spectrometric (ESI-MS) data were in perfect agreement with those reported in the literature [13,15]. The lipid formulation of HoThyRu was prepared through the thin film protocol by co-aggregation with DOTAP (1,2-dioleyl-3-trimethylammoniumpropane chloride). Carefully weighed quantities were dissolved in chloroform and then mixed in the desired DOTAP: HoThyRu 70:30 molar ratio (effective ruthenium complex concentration 30% mol/mol). The resulting solutions were transferred in a round-bottom glass tube, and the solvent was evaporated with anhydrous nitrogen to obtain a homogeneous thin film. Samples were dried under vacuum for at least 24 h to ensure the complete chloroform removal before rehydration with a specific amount of PBS (phosphate-buffered saline pH 7.4, Sigma, Milan, Italy), previously filtered through 0.22 μm filters, so to obtain a total lipid concentration of 1 mM. Finally, samples were vortexed, briefly sonicated, and extruded through polycarbonate membranes with 100 nm sized pores at least 15 times to obtain monodisperse liposome dispersion, as assessed by dynamic light scattering (DLS) control measurements [15]. In detail, the hydrodynamic radius for the HoThyRu/DOTAP nanosystem was found in the 70–100 nm range (Appendix A), which is the typical range of unilamellar vesicles. In addition, PDI close to 1 and zeta potential values of about +40 mV were found, in accordance with previous reports [15].

### 4.2. Cell Cultures

Epithelial-like type human breast adenocarcinoma cells MDA-MB-231 (ATCC, HTB-26^TM^) were grown in DMEM (Invitrogen, Paisley, UK) supplemented with 10% fetal bovine serum (FBS, Cambrex, Verviers, Belgium), L-glutamine (2 mM, Sigma, Milan, Italy), penicillin (100 units/mL, Sigma) and streptomycin (100 μg/mL, Sigma), and cultured in a humidified 5% carbon dioxide atmosphere at 37 °C, according to ATCC recommendations [17]. Human primary adult dermal fibroblasts (HDFa) and human primary epidermal follicular keratinocytes (human hair follicular keratinocytes, HHFKs) were used as control healthy cells, providing ideal cell systems to study toxicological cellular responses [59]. HDFa (ATCC, Manassas, VG, USA) were obtained from the skin of a white male donator (PCS-201-012™), providing an ideal cell system to study toxicological cellular responses. Fibroblasts were cultured in Fibroblast Basal Medium (ATCC) supplemented with recombinant human fibroblast growth factor (rh FGF, 5 ng/mL), L-glutamine (7.5 mM), ascorbic acid (50 µg/mL), hydrocortisone hemisuccinate (1 µg/mL), rh Insulin (5 µg/mL), and Fetal Bovine Serum (FBS, 2%). Moreover, penicillin–streptomycin–amphotericin B solution (penicillin: 10 Units/mL, streptomycin: 10 µg/mL, amphotericin B: 25 ng/mL) was added. HDFa cells were seeded at a density between 2.5–5 × 10^3^ cells/cm^2^ and were passed when approximately 80% to 100% confluence was reached and only if cells were actively proliferating. HHFKs, involved in hair biogenesis, were purchased from ScienCellTM Research Laboratories. Cultures were established by means of specific dissection and dissociation protocols following surgical procedures (human scalp biopsies, ScienCell^®^ # 2440, TAN Record # 944) on appropriate donors based on specific phenotypic requirements (Caucasian race, male gender, age 55 years). HHFK cells were grown in keratinocyte medium (KM, ScienCell^®^ Cat. No. 2101) supplemented with 1% of keratinocyte growth supplement (KGS, ScienCell^®^ Cat. No. 2152) and 1% of penicillin/streptomycin solution (P/S, ScienCell^®^ Cat. No. 0503). All cells were cultured in a humidified 5% carbon dioxide atmosphere at 37 °C, according to the supplier’s recommendations.

### 4.3. Bioscreens In Vitro

The bioactivity of the HoThyRu/DOTAP nanosystem was investigated through the estimation of a “cell survival index” arising from the combination of cell viability evaluation with cell counting, as previously reported by us [11,17]. The cell survival index is calculated as the arithmetic mean between the percentage values derived from the MTT assay and the automated cell count. MDA-MB-231, HDFa, and HHFK cells were inoculated in 96-microwell culture plates at a density of 10^4^ cells/well and allowed to grow for 24 h. The medium was then replaced with fresh medium, and cells were treated for further 48 h with a range of concentrations (1→250 μM) of HoThyRu/DOTAP liposome. Using the same experimental procedure, cell cultures were also incubated with cisplatin (*c*DDP) as a positive control for cytotoxic effects, as well as with the naked AziRu complex as an additional internal control. Cell viability was evaluated via the MTT assay procedure, which measures the level of mitochondrial dehydrogenase activity using the yellow 3-(4,5-dimethyl-2-thiazolyl)-2,5-diphenyl-2H-tetrazolium bromide (MTT, Sigma) as substrate. The assay is based on the redox ability of living mitochondria to convert dissolved MTT into insoluble purple formazan. Briefly, after the treatments, the medium was removed, and the cells were incubated with 20 μL/well of an MTT solution (5 mg/mL) for 1 h in a humidified 5% CO_2_ incubator at 37 °C. The incubation was stopped by removing the MTT solution and by adding 100 μL/well of DMSO to solubilize the obtained formazan. Finally, the absorbance was monitored at 550 nm using a microplate reader (iMark microplate reader, Bio-Rad, Milan, Italy). Cell number was determined by TC20 automated cell counter (Bio-Rad, Milan, Italy), providing an accurate and reproducible total count of cells and a live/dead ratio in one step by a specific dye (trypan blue) exclusion assay. Bio-Rad’s TC20 automated cell counter uses disposable slides, TC20 trypan blue dye (0.4% trypan blue dye *w*/*v* in 0.81% sodium chloride and 0.06% potassium phosphate dibasic solution), and a CCD camera to count cells based on the analyses of captured images. Once the loaded slide is inserted into the slide port, the TC20 automatically focuses on the cells, detects the presence of trypan blue dye, and provides the count. When cells are damaged or dead, trypan blue can enter the cell allowing living cells to be counted. Operationally, after treatments in 96-well culture plates, the medium was removed, and the cells were collected. Ten microliters of cell suspension, mixed with 0.4% trypan blue solution at a 1:1 ratio, were loaded into the chambers of disposable slides. The results are expressed in terms of total cell count (number of cells per mL). If trypan blue is detected, the instrument also accounts for the dilution and shows live cell count and percent viability. Total counts and live/dead ratio from random samples for each cell line were subjected to comparisons with manual hemocytometers in control experiments.

The calculation of the concentration required to inhibit the net increase in the cell number and viability by 50% (IC_50_) is based on plots of data (*n* = 6 for each experiment) and repeated five times (total *n* = 30). IC_50_ values were calculated from a dose-response curve by nonlinear regression using a curve fitting program, GraphPad Prism 8.0, and are expressed as mean values ± SEM (*n* = 30) of five independent experiments.

### 4.4. Colony Formation Assay

MDA-MB-231 cells were seeded out in 100 mm culture Petri dishes at a density of 5 × 10^5^ and incubated for 24 h to allow for attachment to the plate surface. The medium was then replaced with fresh medium, and cells were treated for further 48 h with IC_50_ concentrations of HoThyRu/DOTAP (30 µM, i.e., 9 µM of AziRu) and *c*DDP (7 µM) used as a positive control. After treatments, cells were collected by trypsinization and inoculated in six-well culture plates at a density of 2 × 10^3^ and allowed to grow for about 15 days. On the day of staining, the culture medium was removed, and the cells were washed twice with cold PBS. Then, cells were fixed for 10 min with ice-cold 4% paraformaldehyde, washed twice with cold PBS, and then stained with 0.5% crystal violet (water 40%; methanol 50%; acetic acid 10%; crystal violet 0.5%) in sterile water (500 µL/well) for 30 min at RT. Excess dye was discarded by washing the plates for 15 min in fresh water. After washing, the colonies were counted, and the reported results were representative of three independent experiments.

### 4.5. Fluorescent Detection of Apoptosis, Autophagy, and Necrosis 

Detection of programmed cell death (PCD), i.e., apoptosis and autophagy, as well as the occurrence of necrosis, were evaluated by targeted fluorescent assays. Apoptotic, necrotic, and healthy cells have been monitored by Apoptosis/Necrosis Detection Kit (Abcam, Cambridge, UK, ab176749). DAPI (4′,6-diamidino-2-phenylindole) was used as a blue-fluorescent DNA stain for nuclear regions (DAPI filter, Ex/Em = 350/470 nm). The PS (phosphatidylserine) sensor for early apoptosis uncovering has green fluorescence (FITC filter, Ex/Em = 490/525 nm) upon binding to membrane PS. Loss of plasma membrane integrity for necrosis or late-stage apoptosis detection is associated with the ability of the red fluorescent membrane-impermeable 7-AAD (Cy5 filter, Ex/Em = 546/647 nm) to label the nucleus. For autophagy detection, autophagic vacuoles and autophagic flux in live cells have been monitored with the Autophagy Assay Kit (Abcam, ab139484), based on a cationic amphiphilic tracer (CAT) dye with spectral characteristics similar to FITC that rapidly and selectively labels autophagic vacuoles. The CAT dye has been optimized for both minimal staining of lysosomes, and bright fluorescence upon incorporation into pre-autophagosomes, autophagosomes, and autolysosomes (autophagolysosomes), allowing for a rapid and quantitative approach to monitoring autophagy in live cells (FITC filter, Ex/Em = 490/525 nm). Rapamycin (10 µM) was used as a positive control because of being an autophagy inducer [17]. MDA-MB-231 cells were grown in a black wall/clear bottom 96-well microplate (Corning Life Sciences, Bedford, MA, USA) and treated with IC_50_ of HoThyRu/DOTAP (30 µM, i.e., 9 µM of AziRu) for 48 h. According to kit assay protocols, appropriate quantities of reagents were added to the cells. After incubation at room temperature for 30 min (protected from light), cells were washed twice with assay buffer, and then fluorescence intensity was monitored at the indicated wavelengths using a confocal microscope (Zeiss LSM 900 Airyscan 2) at 40× magnification (oil immersion objective lens). Fluorescence intensity for apoptosis and autophagy activation was determined with ImageJ FIJI software 2.9. Specifically, we selected and analyzed different and randomized areas. The outputted numbers of all the images were then analyzed and graphed. Optimal parameter settings were found using both the green-fluorescent PS sensor-positive controls and CAT dye-positive controls. 

### 4.6. Transwell Invasion and Migration Assay

Cell Invasion Assay (Collagen I, 24-well, 8 μm—ab235887) was used for the Transwell-like cell invasion and migration assay following the manufacturer’s recommendations (Abcam, Cambridge, UK). These assays employ a Boyden chamber coated with an 8 μm of Collagen I membrane. Since cell invasion and migration can be influenced by culture medium, preliminary experiments were performed to optimize the assay by decreasing serum concentration (serum starvation). Briefly, the top chamber was coated with 100 μL of Collagen I at 37 °C in a CO_2_ incubator for 3 h. Then, the coated plate was washed three times, and 2.3 × 10^5^ cells, previously treated or not for 48 h with a sub-IC_50_ concentration of HoThyRu/DOTAP (24 μM, i.e., 7.2 µM of AziRu), were plated in serum-free medium. The bottom chamber was filled with FBS containing medium or control invasion inducer (provided by the kit for the positive control) and incubated at 37 °C with 5% CO_2_ for 48 h. After washing the cells in the bottom chamber, they were labeled by Cell Dye (Abcam), and the measure of the number of cells invaded or migrated was performed using a microplate reader (iMark microplate reader, Bio-Rad, Milan, Italy) at 485 nm excitation and 530 nm emission. All the experiments were independently repeated at least three times. 

### 4.7. In Vitro Wound-Healing Assay

Cell collective migration ability was determined by the wound healing assay, an established two-dimensional (2D) technique also known as the scratch assay. MDA-MB-231 cells, previously treated or not for 48 h with a sub-IC_50_ concentration of HoThyRu/DOTAP (24 μM, i.e., 7.2 µM of AziRu), were then cultured into 24-well plates at a density of 1.5 × 10^5^ in serum-free medium. The bottom of each well was scratched with a sterile pipette tip (P10 micropipette tip). Vertical scratches were drawn through an about 80% confluent monolayer. The culture medium was removed, and cells were washed twice with sterile PBS to remove cell debris. Since cell proliferation can compete with cell migration to fill the gap made during the assay, preliminary experiments were performed to optimize the medium by decreasing serum concentration (serum starvation) to control cell proliferation. At the established time endpoints, migration was monitored under a phase contrast microscope. The wound area and the percentage of wound closure depending on cell migration ability were determined by ImageJ FIJI software. Independent experiments were performed three times. 

### 4.8. RT-qPCR

RNA extraction from untreated MDA-MB-231 (control cells) and MDA-MB-231 treated with HoThyRu/DOTAP at a sub-IC_50_ concentration (24 µM) for 48 h was performed with NORGEN Total RNA Purification Kit (Cat. 17200) according to the manufacturer protocol. Total cDNA was obtained using Applied Biosystems High-Capacity cDNA Reverse Transcription Kit with RNAse Inhibitors (Cat. 4374966). RT-qPCR was performed using Applied Biosystems FAST SYBR Green Master Mix (Cat. 4385612) and specific qPCR primers for E-Cadherin (Epithelial Cadherin), N-Cadherin (Neural Cadherin), Vimentin, Slug (Snail2), Snail (Snail1), and GAPDH (used as housekeeping gene) obtained by Eurofins Scientifics. The primer sequences are listed below. E-cadherin: FW-GAG TGC CAA CTG GAC CAT TCA GTA; RV-AGT CAC CCA CCT CTA AGG CCA TC. N-cadherin: FW-GAC ATT GTC ACT GTT GTG TCA CCT G; RV-CCG TGC CTG TTA ATC CAA CAT C. Vimentin: FW-TGA CAA TGC GTC TCT GGC AC; RV-CCT GGA TTT CCT CTT CGT. Slug: FW-TTT CTT GCC CTC ACT GCA AC; RV-ACA GCA GCC AGA TTC CTC AT. Snail: FW-CCT CCC TGT CAG ATG AGG AC; RV-CTT TCG AGC CTG GAG ATC CT. GAPDH: FW-GGA GTC AAC GGA TTT GGT CG; RV-CTT CCC GTT CTC AGC CTT GA. Qiagen Rotor-Gene Q MDx Platform and 0.2 µL tubes and caps were used for the RT-qPCR run, the results were then analyzed using the 2^−∆∆Ct^ method. Graphs and *t*-test analyses were conducted using GraphPad Prism 8.0. 

### 4.9. Subcellular Fractionation 

MDA-MB-231 cells were grown on standard 100 mm culture dishes by plating 8 × 10^5^ cells. After 24 h of growth, the cells were incubated with an IC_50_ concentration of HoThyRu/DOTAP (30 µM) for 24 h under the same experimental conditions described for bioscreen assays. At the end of the treatment, the culture medium was collected, and the cells were enzymatically harvested by trypsin, then centrifuged at RT for 3 min at 300× *g*. The cell pellets were resuspended in 500 μL of a solution I (10 mM HEPES pH 7.9, 10 mM KCl, 0.1 mM MgCl_2_, 0.1 mM EDTA, 0.1 mM DTT, Protease Inhibitor Cocktail) and centrifuged at 2000 rpm for 10 min at 4 °C. The supernatant, representing the cytosolic fraction, was separated from the pellets, which, in turn, contained the nuclear and mitochondrial fractions. The pellets were washed three times with solution I, and then 200 µL of lysis buffer (10 mM HEPES, 3 mM MgCl_2_, 40 mM KCl, 5% glycerol, 1 mM DTT, 0.2% NP40) was added and incubated for 30 min in ice. After centrifugation at 4 °C for 30 min at 500× *g*, the pellets enriched in the nuclear fraction were obtained. To acquire the purified DNA fraction, the pellets were suspended in DNA lysis buffer (50 mM Tris-HCl, pH 8.0, 0.5 mM EDTA, 100 mM NaCl, 1% SDS, 0.5 mg/mL proteinase K) and incubated at 50 °C for 1 h. After incubation, 10 mg/mL RNase was added to the lysates and incubated for 1 h at 50 °C. DNA was precipitated with NaOAc pH 5.2 and ice-cold 100% EtOH and then centrifuged at 14,000× *g* for 10 min. The pellets were dissolved in TE buffer (10 mM Tris-HCl, pH 8.0, 1 mM EDTA). To obtain the mitochondrial fraction, the reagent-based method of mitochondria isolation kit for mammalian cells (Thermo Scientific^TM^, Waltham, MA, USA) was used. Briefly, 2 × 10^7^ cells treated as above reported, were centrifugated (850× *g* for 2 min), and the pellet was suspended in 800 µL of mitochondria isolation reagent A for max 2 min; 10 µL of mitochondria isolation reagent B was then added. Tubes were incubated on ice for 5 min, after which 800 µL of mitochondria isolation reagent C was added. Finally, differential centrifugation protocols to separate the mitochondrial and cytosolic fractions with a microcentrifuge (Eppendorf, Tokyo, Japan) were performed according to the manufacturer’s datasheet. Aliquots of culture media, cellular pellets, cytosolic, mitochondrial, and nuclear fractions, as well as DNA samples, were subsequently analyzed by inductively coupled plasma-mass spectrometry (ICP-MS) to determine the ruthenium amounts in each sample. 

### 4.10. Animals and Experimental Design

Four-week-old female athymic nude Foxn1^nu^ mice (23–26 g) were purchased from Envigo RMS (Udine, Italy) and kept in an animal care facility at a controlled temperature of 22 ± 3 °C, humidity (50 ± 20%), and on a 12:12 h light-dark cycle (lights on at 07:00 h). All mice were acclimatized to the environmental conditions for at least five days before starting the xenograft experiments. They were housed in Plexiglas cages (five mice/cage) equipped with air lids, kept in laminar airflow hoods, and maintained under pathogen-limiting conditions. Animals were maintained with free access to sterile food and water. Sterile food was purchased from Envigo (Teklad global 18% protein #2018SX, Envigo, Madison, WI, USA). Cages and water were autoclaved before use. Mice were randomly divided into two groups (untreated xenotransplanted and xenotransplanted treated with HoThyRu/DOTAP) and then used to set up xenograft models (*n* = 5 animals for each experimental group). Animal studies were conducted in accordance with the guidelines and policies of the European Communities Council and were approved by the Italian Ministry of Health (n.354/2015-PR). Protocols and procedures for in vivo studies were performed under the supervision of veterinary experts according to European Legislation. All procedures were carried out to minimize the number of animals used and their suffering.

### 4.11. Generation of Human TNBC-Derived Xenograft Models in Nude Mice

At 80% confluence, MDA-MB-231 cells were trypsinized and harvested. Cell number was determined by TC20 automated cell counter (Bio-Rad, Milan, Italy) with a specific dye (trypan blue) exclusion assay. Aliquots containing 5 × 10^6^ cells were opportunely 1:3 mixed in Matrigel^®^ Matrix (Growth Factor Reduced, Corning, Bedford, MA, USA), and tumors were established by subcutaneous (s.c.) injection into the right flank of each mouse. Mice were randomly assigned to each of the two xenotransplanted experimental groups (untreated xenotransplanted and xenotransplanted treated). 

### 4.12. Treatments In Vivo: Experimental Protocols and Therapeutic Scheme

HoThyRu/DOTAP treatment in vivo started two weeks post-tumor implant by intraperitoneal (i.p.) injection, according to a standardized and tested protocol (always subordinate to the check of the actual development of the tumors) [18]. In brief, 15 mg/kg of HoThyRu/DOTA, contained in 300 μL of sterile water (molecular biology grade water, Corning, Bedford, MA, USA), were administered to the treated mice group once a week for 28 days (4 weeks). The control group (untreated xenotransplanted) received an equal volume of sterile PBS. After 28 days of the administration, animals were sacrificed, and tumors and organs were first appropriately collected and then carefully weighed and photographed. All experimental procedures were carried out in compliance with international and national law and policies (EU Directive 2010/63/EU for animal experiments, ARRIVE guidelines, and the Basel declaration, including the 3R concept) and approved by the Italian Ministry of Health (n.354/2015-PR).

### 4.13. Tumor Volume Determination by Caliper Measurements 

Starting from the week-later implantation of human BCC in nude mice (measurable subcutaneous tumors of about 300–500 mm^3^), tumor volumes in xenotransplanted mice were determined throughout the study using an external caliper. Specifically, the largest longitudinal (length) and transverse (width) diameters were monitored and recorded every two days. Tumor volume measurements were then calculated by the formula V = (Lenght × Width^2^)/2.

### 4.14. Animal Supervision and Monitoring throughout the Preclinical Study

Animals were checked daily by veterinarians, and their state of health was monitored continuously. Mice body weights were recorded every two days by MS-Analytical and Precision Balance (Mettler Toledo, Columbus, OH, USA). Special attention was given to the tumor size as well as to the skin area near the tumor lesion to avoid animal pain. 

### 4.15. Surgical Procedures, Harvest of Tumors, and Biological Samples Collection

At the end of the study (28 days of treatment, once a week, with HoThyRu/DOTAP, as described above), mice (untreated xenotransplanted group and xenotransplanted treated with HoThyRu/DOTAP group) were sacrificed in a chamber containing CO_2_ according to AVMA guidelines for the euthanasia of animals. Any effort was made to minimize animal pain and discomfort. Tumors, organs, and tissues (blood, heart, liver, kidneys, brain, spleen, and lungs) were meticulously collected by surgical procedures under strictly aseptic conditions following sacrifices. All the animal experiments were conducted according to the guidelines of the Institutional Animal Care and Use Committee (IACUC). After macroscopic evaluations, biological samples were cataloged and properly cryopreserved at −80 °C until analysis.

### 4.16. Ruthenium Bioaccumulation by Inductively Coupled Mass Spectrometry (ICP-MS)

Spectrometry (ICP-MS) was used for a highly sensitive determination of ruthenium distribution in MDA-MB-231 cells subfractions after incubation for 24 h with the HoThyRu/DOTAP nanosystem, as well as for ruthenium determination in tissue samples from animals exposed to HoThyRu/DOTAP regimens. Biological samples were subjected to oxidative acid digestion with a mixture of 69% nitric acid and 30% *v*/*v* hydrogen peroxide in an 8:1 ratio, using high temperature and pressure, under a microwave-assisted process. A proper dilution was made, and the suspension obtained for each sample was introduced to the plasma. The mineralized samples were recovered with ultrapure water and filtered using 0.45 μm filters. The determination of ruthenium was carried out on Inductively Coupled Plasma Mass Spectrometry (ICP-MS) instrument Aurora M90 Bruker. The quantitative analysis was performed via the external calibration curve method. In the analyzed samples and fractions, the ruthenium content is expressed as a percentage of the total ruthenium administered throughout in vitro incubations following a standardized protocol.

### 4.17. Statistical Data Analysis

All data were presented as mean values ± SEM. Statistical analysis was performed using one-way or two-way ANOVA followed by Dunnett’s or Bonferroni’s for multiple comparisons. GraphPad Prism 8.0 software was used for analysis. Differences between means were considered statistically significant when *p* ≤ 0.05 was achieved. The sample size was chosen to ensure alpha 0.05 and power 0.8. Animal weight was used for randomization and group allocation to reduce unwanted sources of variations by data normalization. No animals and related ex vivo samples were excluded from the analysis. In vivo and in vitro studies were carried out to generate groups of equal size, using randomization and blinded analysis.

## Figures and Tables

**Figure 1 ijms-24-06473-f001:**
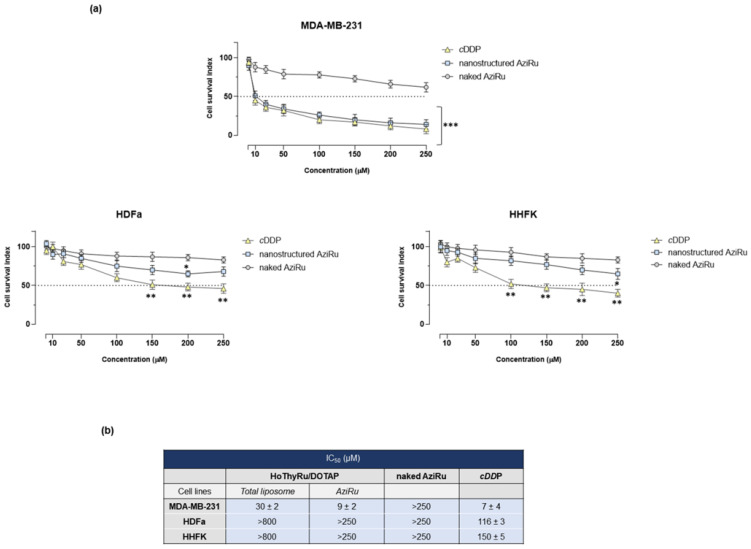
Biological effects of the HoThyRu/DOTAP nanosystem in TNBC cells and in healthy cultures. (**a**) Cell survival index, evaluated by the MTT assay and live/dead cell ratio analysis, for TNBC MDA-MB-231 cells and for healthy primary dermal fibroblasts (HDFa) and primary epidermal follicular keratinocytes (HHFKs) following 48 h of incubation with the indicated concentration (range 1→250 µM) of AziRu loaded in the HoThyRu/DOTAP nanosystem (nanostructured AziRu) and the naked AziRu. In the same experimental conditions, cisplatin (cDPP) is used as the reference drug. Data in line graphs are expressed as percentages of untreated control cells and are reported as mean of five independent experiments ± SEM (*n* = 30). * *p* ˂ 0.05 vs. control cells; ** *p* < 0.01 vs. control cells; *** *p* < 0.001 vs. control cells. (**b**) IC_50_ values (µM) of the HoThyRu/DOTAP liposomal formulation, the actual Ru(III) complex (AziRu) in the nanosystem, the naked AziRu complex, and cisplatin (cDDP) in the tested cell lines after 48 h of incubation in vitro. The AziRu IC_50_ value corresponds to the effective ruthenium complex concentration (30% mol/mol) carried by the HoThyRu/DOTAP nanoformulation. IC_50_ values are reported as mean ± SEM (*n* = 30).

**Figure 2 ijms-24-06473-f002:**
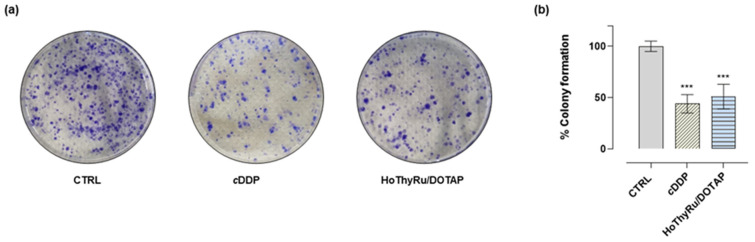
Colony formation assay in the experimental model of TNBC. (**a**) Representative images of MDA-MB-231 cells stained with 0.5% crystal violet at the experiment endpoint. Cells were treated or not (Ctrl) with IC_50_ concentrations of HoThyRu/DOTAP and cisplatin (*c*DDP), as indicated in the experimental section. *c*DDP is used as a cytotoxic reference drug. (**b**) Quantification by bar graphs of the cell colonies formation after the indicated treatments. *** *p* < 0.001 vs. untreated cells (Ctrl).

**Figure 3 ijms-24-06473-f003:**
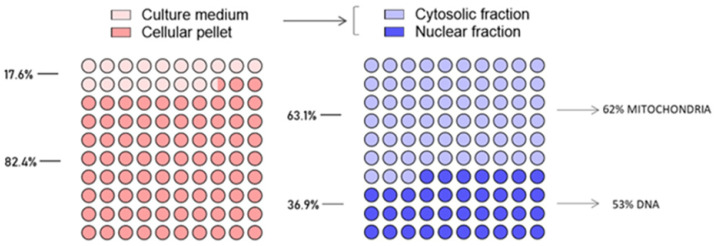
Intracellular ruthenium(III) complex bioaccumulation after HoThyRu/DOTAP application to MDA-MB-231 cells. Inductively coupled plasma-mass spectrometry (ICP-MS) for the analysis of ruthenium distribution between MDA-MB-231 cells and culture media after incubation for 24 h with the IC_50_ concentration of HoThyRu/DOTAP, as well as intracellular ruthenium accumulation following cellular uptake and subcellular fractionation. In the reported fractions, ruthenium content is expressed as percentage of the total ruthenium administered during incubations in vitro. Results were derived from the average values of three independent experiments.

**Figure 4 ijms-24-06473-f004:**
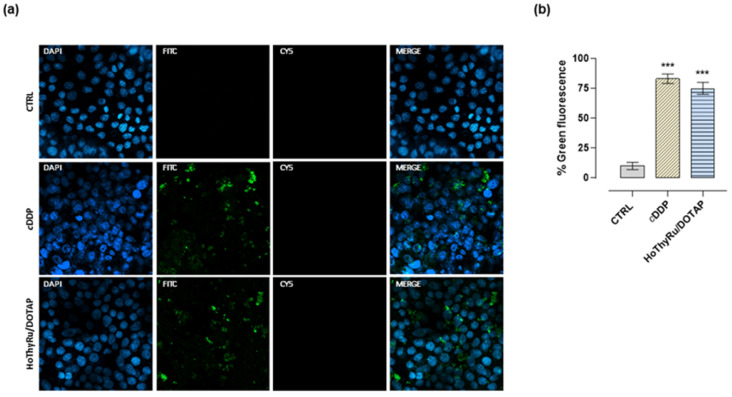
Apoptosis activation in MDA-MB-231 cells by confocal microscopy in response to HoThyRu/DOTAP treatment. (**a**) Apoptotic, necrotic, and healthy cells have been monitored by confocal microscopy after incubation for 48 h with IC_50_ concentrations of HoThyRu/DOTAP and cisplatin (cytotoxic positive control). Nuclei emit blue fluorescence (blue nuclear stain, DAPI filter, Ex/Em = 350/470 nm). Apoptotic cells have green fluorescence (FITC filter, Ex/Em = 490/525 nm) upon binding to membrane PS (phosphatidylserine). Necrotic cells are associated with nuclear red fluorescence (Cy5 filter, Ex/Em = 546/647 nm). In merged images (Merge), the fluorescent patterns from cell monolayers are overlapped. Fluorescent microphotographs (40× oil immersion objective lens) are representative of three independent experiments. (**b**) Percentage of Green Detection Reagent-positive MDA-MB-231 cells following the indicated treatments in vitro with respect to untreated control cells. *** *p* < 0.001 vs. control cells.

**Figure 5 ijms-24-06473-f005:**
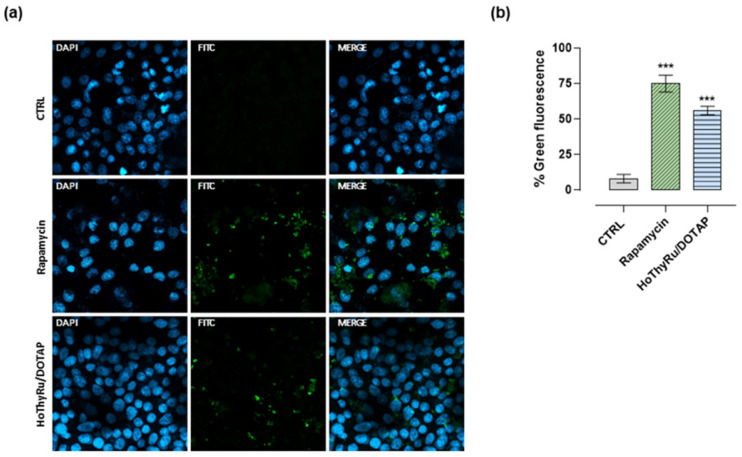
Autophagy fluorescent detection in MDA-MB-231 cells treated with HoThyRu/DOTAP. (**a**) Autophagy detection by confocal microscopy showing nuclei (blue nuclear stain, DAPI filter, Ex/Em = 350/470 nm) and autophagic vesicles (green fluorescence signal, FITC filter, Ex/Em = 490/525 nm) in control MDA-MB-231 cells (Ctrl), or in cells treated with 10 µM Rapamycin for 48 h, and with IC_50_ of HoThyRu/DOTAP for 48 h. In merged images (Merge), the fluorescent patterns from cell monolayers are overlapped. The shown microphotographs (40× oil immersion objective lens) are representative of three independent experiments. (**b**) Percentage of Green Detection Reagent-positive MDA-MB-231 cells following the indicated treatments in vitro with respect to untreated control cells. *** *p* < 0.001 vs. control cells.

**Figure 6 ijms-24-06473-f006:**
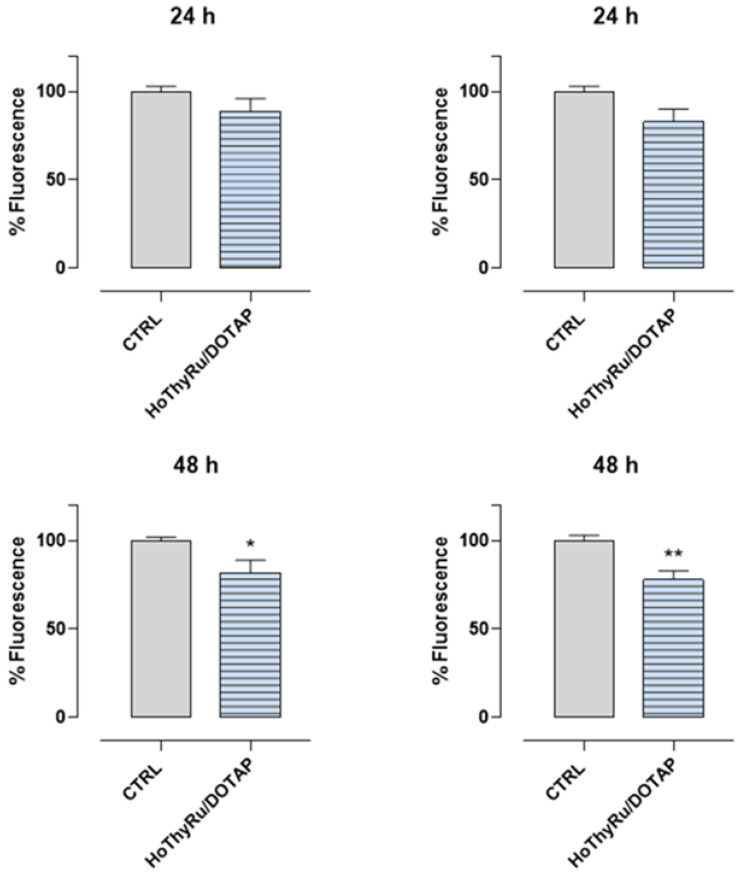
Invasion and migration ability of MDA-MB-231 cells in response to HoThyRu/DOTAP treatment. MDA-MB-231 cells were starved and treated or not with a sub-IC_50_ concentration of HoThyRu/DOTAP (24 μM, i.e., 7.2 µM of AziRu) for the indicated times (24 and 48 h). The ability of cells to invade the matrix and then migrate through a semipermeable membrane in the Boyden chamber in response to HoThyRu/DOTAP application in vitro was analysed directly in fluorescence according to the manufacturer’s recommendations and reported in bar graphs. Data originate from the average ± SEM values of three independent experiments. * *p* ˂ 0.05 vs. control cells; ** *p* < 0.01 vs. control cells.

**Figure 7 ijms-24-06473-f007:**
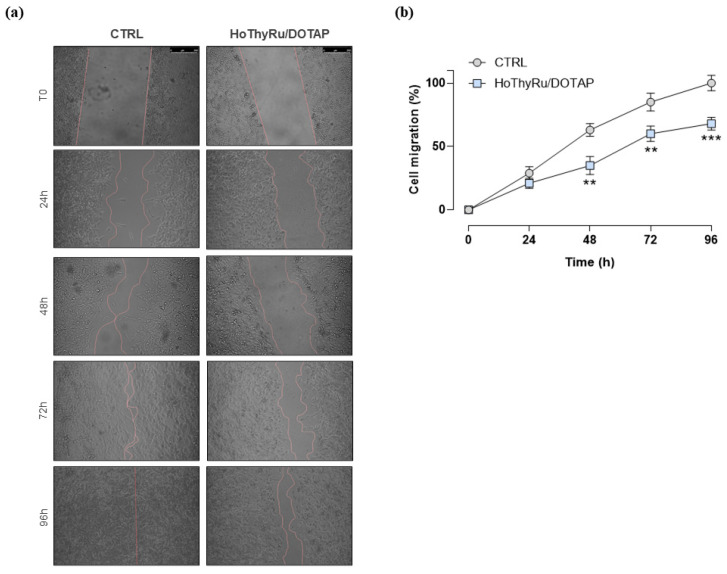
Wound healing assay showed inhibitory effects of HoThyRu/DOTAP on cell migration. (**a**) Representative images by light microscopy showing MDA-MB-231 cell migration for the indicated times (0, 24, 48, 72, and 96 h), previously treated or not for 48 h with HoThyRu/DOTAP at the sub-IC_50_ concentration of 24 µM. The scale bar represents 250 µM. (**b**) At the endpoints, migration was monitored under a phase contrast microscope (10× objective), and the percentage of wound closure depending on cell migration ability was determined by ImageJ FIJI software and reported in a line graph as the average ± SEM values of three independent experiments. ** *p* < 0.01 vs. control cells; *** *p* < 0.001 vs. control cells.

**Figure 8 ijms-24-06473-f008:**
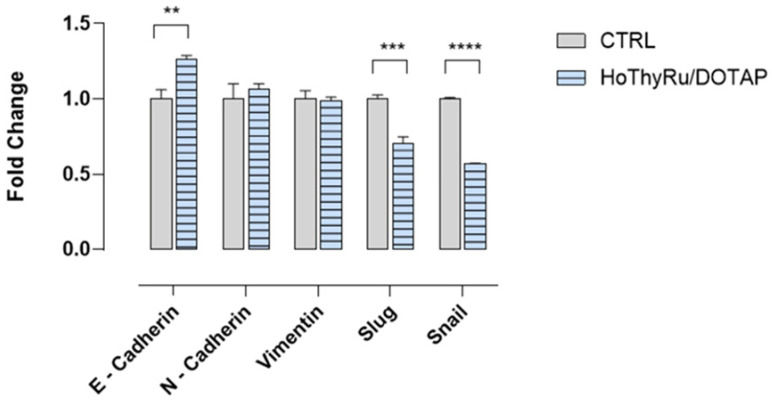
Expression analysis of a limited panel of EMT genes by RT-qPCR following HoThyRu/DOTAP application in vitro. RT-qPCR analysis of the EMT pathway genes E-cadherin, N-cadherin, vimentin, Slug, and Snail, performed on MDA-MB-231 cells treated or not with HoThyRu/DOTAP for 48 h. The mRNA expression levels of each gene were normalized using the GAPDH as a housekeeping gene and are indicated as the fold change with respect to untreated control cultures. Values represent the mean ± SEM of three independent experiments, each performed in duplicate. ** *p* < 0.01 vs. control cells; *** *p* < 0.001 vs. control cells; **** *p* < 0.0001 vs. control cells. The report of RT-qPCR analysis is shown in Appendix A.

**Figure 9 ijms-24-06473-f009:**
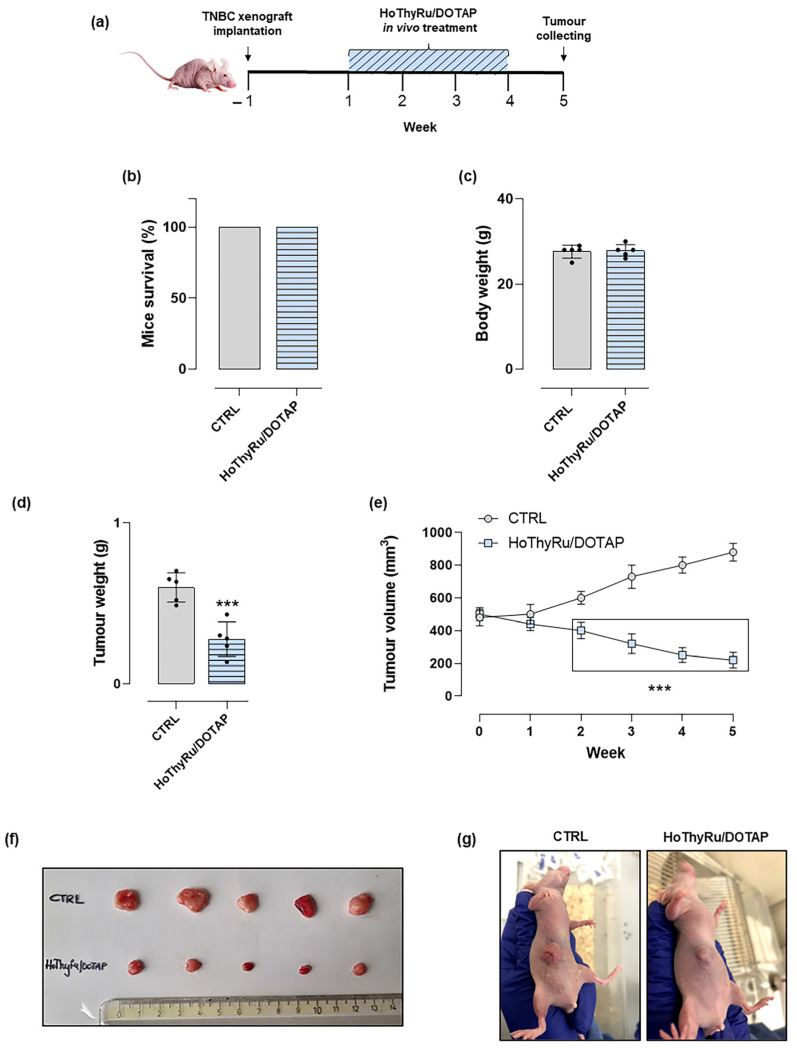
Animal biological responses to HoThyRu/DOTAP administration in vivo. (**a**) Experimental protocol and therapeutic scheme based on intraperitoneal (i.p.) administrations of HoThyRu/DOTAP (15 mg/kg) once a week for 28 days. (**b**) Overall mice survival and (**c**) body weights at the end of the study (5 weeks from the start of treatments). Control group (untreated xenotransplanted, *n* = 5 animals); xenotransplanted treated group (HoThyRu/DOTAP, *n* = 5 animals). (**d**) Weight analysis of the explanted tumor masses at the end of the study and (**e**) tumour volumes evaluation over time throughout in vivo experiments. Control group (untreated xenotransplanted, *n* = 5 animals); xenotransplanted treated group (HoThyRu/DOTAP, *n* = 5 animals). (**f**) Explanted tumor masses at the end point of the study from untreated (Control) and treated (HoThyRu/DOTAP) xenotransplanted animal groups. (**g**) Representative animal photographs at the end of the preclinical study relating to untreated xenotransplanted mice (Control) and treated xenotransplanted mice (HoThyRu/DOTAP) showing tumour inhibition by HoThyRu/DOTAP administration. Statistical analysis was conducted by one-way ANOVA followed by Bonferroni’s for multiple comparisons. *** *p* ˂ 0.001 vs. control animal group.

**Figure 10 ijms-24-06473-f010:**
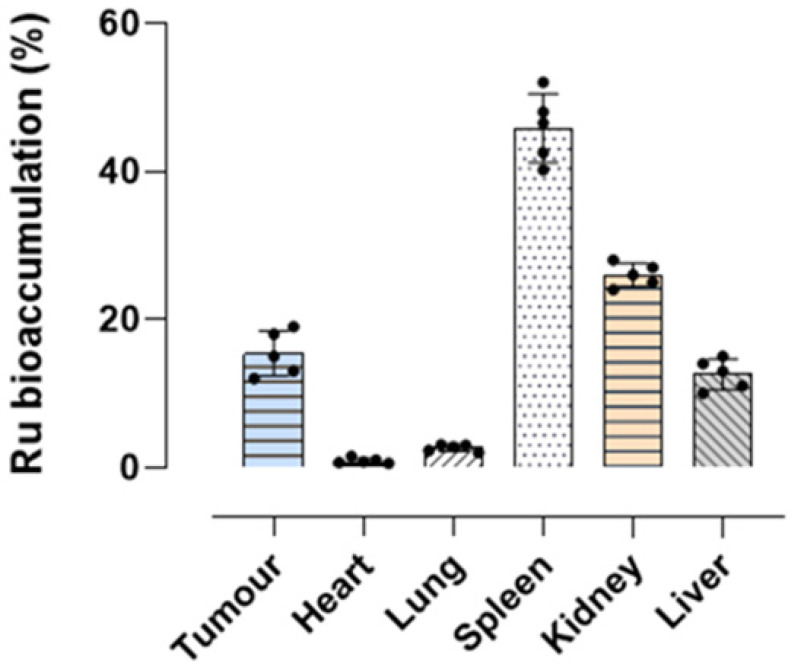
Ruthenium bioaccumulation in mice after HoThyRu/DOTAP regimen in vivo. Percentage of ruthenium amounts revealed by ICP-MS analyses and plotted in bar graph for the indicated body districts (heart, lung, spleen, kidney, and liver), including tumour lesions, at the endpoint (4 weeks) of the preclinical study. After weekly administrations of HoThyRu/DOTAP (15 mg/kg, i.p., once a week for 4 weeks), the mice were sacrificed, and organs and tissues were appropriately collected to analyze the ruthenium content (*n* = 5 animals).

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
