# Peer review of "Triple Negative Breast Cancer Preclinical Therapeutic Management by a Cationic Ruthenium-Based Nucleolipid Nanosystem"

_ijms, 2023, doi:10.3390/ijms24076473_

Round 1

Reviewer 1 Report

The submitted manuscript evaluated the biological activity of a cationic  liposomal ruthenium(III) complex AziRu-containing nanoformulation aiming at a new treatment for triple negative breast cancer. Introduction of the manuscript presents an informative and updated literature review, relevant to the topic and aim of work. Results and discussion are also adequate. Below are some suggestions, comments, and inquiries needing clarification:

1. The authors could provide a figure illustrating the nanosystem. That would enrich the manuscript.

2. Authors should provide information on DLS data, such as size, PDI and Zeta potential. Also, information on the pH of the formulation is relevant as treatment is carried on by parenteral administration.

3. Information on formulation's stability is also needed: is the nanoformulation stable at room temperature? For how long? 

4. Authors should justify the reason for intraperitoneal administration and for the 15 mg/kg dose. 

5. Minor gramatical corrections must be made, such as in "exsisting" (line 19), "BBC" instead of BCC (line 68), "chemioresistance" in line 89 and "cytosl" in line 444.

Author Response

Response to Decision letter for ijms-2303051 (article)

“Triple Negative Breast Cancer Preclinical Therapeutic Management by a Cationic Ruthenium-Based Nucleolipid Nanosystem”

Section - Biochemistry

Metal-Based Complexes in Cancer 2.0

Point-by-point author's response to reviewers.

The manuscript by Ferraro et al. has considerably benefited according to the reviewers' reports. Their suggestions provided the opportunity to further improve the research article. Therefore, we thank the reviewers for their valuable recommendations and suggestions. In what follows we reply to their comments.

Review Report

Reviewer #1:

The submitted manuscript evaluated the biological activity of a cationic liposomal ruthenium (III) complex AziRu-containing nanoformulation aiming at a new treatment for triple negative breast cancer. Introduction of the manuscript presents an informative and updated literature review, relevant to the topic and aim of work. Results and discussion are also adequate. Below are some suggestions, comments, and inquiries needing clarification.

Author's Reply to the Reviewer#1

We thank the reviewer for his overall positive feedback. Below are our responses to his suggestions.

  1. The authors could provide a figure illustrating the nanosystem. That would enrich the manuscript.

We fully agree with the reviewer, and we added a new figure in supplementary documents concerning the nanosystem structural details (Figure S1). Thus, we have renumbered supplementary figures included in manuscript.

  1. Authors should provide information on DLS data, such as size, PDI and Zeta potential. Also, information on the pH of the formulation is relevant as treatment is carried on by parenteral administration.

We thank the reviewer for raising this point, since it is an aspect that we particularly care about. Indeed, the manuscript is full of bibliographic references relating to our previous papers which focus on the formulation, structural details, and physico-chemical characteristics of the HoThyRu/DOTAP nanosystem. Among these, ref. 15 (Biomacromolecules 2013; 14(8): 2549–2560; doi: 10.1021/bm400104b) contains all the structural information about the HoThyRu/DOTAP nanosystem. For the experiments reported in this paper, as per practice liposomes were subjected to DLS control prior to the use in preclinical studies. In detail, size in the 70-100 nm range, PDI close to 1 and zeta potential values of about +40 mV were found. Following reviewer’s suggestion, these details are now included in the manuscript within the experimental section (HoThyRu/DOTAP liposome preparation, line 119-122), along with a new figure provided as supplementary material (Figure S3) and reporting the hydrodynamic radius distribution function obtained by DLS analysis performed on the HoThyRu/DOTAP nanoformulation. Concerning the pH of the nanoformulation, as readily reported in the experimental section, liposomes were prepared using PBS (Phosphate Buffered Saline pH 7.4, Sigma, Milan, Italy), thereby providing the final pH of the saline solution containing the formulations.

  1. Information on formulation's stability is also needed: is the nanoformulation stable at room temperature? For how long?

As for the previous data, this information is already provided in our former reports (in particular, for the HoThyRu/DOTAP see ref. 15, Biomacromolecules 2013; 14(8): 2549–2560; doi: 10.1021/bm400104b). The stability of the HoThyRu/DOTAP nanoformulation at room temperature was assessed by combining several techniques and was proved to be on the order of months. Therefore, we are talking about liposomal nanosystems stable for a long time under bench conditions. First, we made visual inspections of the glass cuvette containing the liposomes, which did not show any detectable sediment also after long storage, nor significant variations in appearance and colour. Then, both UV and DLS measurements over time were performed to follow UV spectral and size or polydispersity changes, respectively. In any cases, no significant variation was found. Finally, electron paramagnetic resonance (EPR) spectra, performed on the same samples after three months, showed no variation of the signals, confirming the stability with time of the bilayers formed by the DOTAP nanoaggregates hosting the HoThyRu complex.

  1. Authors should justify the reason for intraperitoneal administration and for the 15 mg/kg dose.

The experimental protocol (dose and administration route) that we used for in vivo treatments by the HoThyRu/DOTAP nanoformulation derives from literature studies on corresponding drugs based on transition metals (in particular, cisplatin and congeners). One of the reference papers can be Johnsson et al., Cancer Chemother Pharmacol (1995) 37: 23-31. Following these indications, HoThyRu/DOTAP was administered at a dose which allows to deliver an amount of the active AziRu complex corresponding to that of cisplatin. Following this path, we set up a proposal for animal experimentation that was approved by the Italian Ministry of Health, as reported in the manuscript. Moving in this direction, we  performed preliminary experiments to validate the experimental procedure, then used for the in vivo studies reported in our previous works (ref. 17, Piccolo et al., Sci Rep. 2019;9(1):7006. doi: 10.1038/s41598-019-43411-3; ref. 18, Piccolo et al., Cancers. 2021;13(20):5164. doi: 10.3390/cancers13205164). Further details can be found in the experimental section (Treatments in vivo: experimental protocols and therapeutic scheme).

  1. Minor gramatical corrections must be made, such as in "exsisting" (line 19), "BBC" instead of BCC (line 68), "chemioresistance" in line 89 and "cytosl" in line 444.
  2. Thanks for highlighting these corrections.

Reviewer 2 Report

In this manuscript, the authors synthesized a nanosystem composed of a ruthenium complex and cationic lipid (HoThyRu/DOTAP) and investigated its efficacy against triple-negative breast cancer cells (MDA-MB-231) both in vitro and in vivo. In the in vitro experiments, the authors demonstrated that the HoThyRu/DOTAP nanosystem effectively suppressed the proliferation, colony formation, migration, and invasion of tumor cells. Furthermore, the nanosystem induced apoptosis and autophagy in the cancer cells. Moreover, in vivo studies demonstrated that the nanosystem reduced tumor mass without affecting the body weight of the mice. Overall, the manuscript presents the use of HoThyRu/DOTAP as a potential therapeutic agent for triple-negative breast cancer. Comments and suggestions regarding the manuscript are described below.

 1.     It would benefit the readers if the authors included a schematic diagram illustrating the composition of their nanosystem and the mechanisms by which it induces cell death.

2.     It is suggested that the authors provide a brief description of the anti-cancer molecular mechanisms of the Ru complex in the "Introduction" section.

3.     Figure 1: The authors only presented the cell survival index data for the liposomal formulation of Ru (HoThyRu/DOTAP). It is recommended that the authors include information regarding the cell survival index when cells are treated with the Ru complex alone (AziRu).

4.     Figure 8: In addition to analyzing the mRNA levels of E-cadherin, N-cadherin, vimentin, Slug, and Snail, it is recommended that the authors also investigate the protein expression after treatment with HoThyRu/DOTAP.

5.     Figure 10: Does the accumulation of nanoformulation in the kidney and liver cause any toxicities in these tissues?

6.     A typo is found: Title “Ruthe-nium”

Author Response

Response to Decision letter for ijms-2303051 (article)

“Triple Negative Breast Cancer Preclinical Therapeutic Management by a Cationic Ruthenium-Based Nucleolipid Nanosystem”

Section - Biochemistry

Metal-Based Complexes in Cancer 2.0

Point-by-point author's response to reviewers.

The manuscript by Ferraro et al. has considerably benefited according to the reviewers' reports. Their suggestions provided the opportunity to further improve the research article. Therefore, we thank the reviewers for their valuable recommendations and suggestions. In what follows we reply to their comments.

Review Report

Reviewer #2:

In this manuscript, the authors synthesized a nanosystem composed of a ruthenium complex and cationic lipid (HoThyRu/DOTAP) and investigated its efficacy against triple-negative breast cancer cells (MDA-MB-231) both in vitro and in vivo. In the in vitro experiments, the authors demonstrated that the HoThyRu/DOTAP nanosystem effectively suppressed the proliferation, colony formation, migration, and invasion of tumor cells. Furthermore, the nanosystem induced apoptosis and autophagy in the cancer cells. Moreover, in vivo studies demonstrated that the nanosystem reduced tumor mass without affecting the body weight of the mice. Overall, the manuscript presents the use of HoThyRu/DOTAP as a potential therapeutic agent for triple-negative breast cancer. Comments and suggestions regarding the manuscript are described below.

Author's Reply to the Reviewer#2

First, we thank the reviewer for his valuable suggestions which allowed to further improve the quality and the impact of the manuscript. Below please find our responses to criticisms.

  1. It would benefit the readers if the authors included a schematic diagram illustrating the composition of their nanosystem and the mechanisms by which it induces cell death.

We absolutely agree with the reviewer on this point. Thus, a new figure was included in the manuscript as supplementary material (Figure S1) focusing on structural details of the HoThyRu/DOTAP nanosystem and its multimodal mechanisms of action in cancer cells. Of course, supplementary figures provided within the manuscript have been renumbered. Concerning additional information on the anticancer mechanism of action in BCC, this concern is reviewed in point 2.  

  1. It is suggested that the authors provide a brief description of the anti-cancer molecular mechanisms of the Ru complex in the "Introduction" section.

In line with the reviewer’s suggestion, the introduction section of the manuscript has been further improved by including additional data concerning the anticancer mechanism of action of HoThyRu/DOTAP in breast cancer cells (BCCs). Thus, we have now expanded the introduction section in the revised manuscript (line 68-71). However, this section already comprises information on the HoThyRu/DOTAP mechanism of action by activation of PCD pathways (both apoptosis and autophagy), with related references to our previous papers focusing on these topics (ref. 8, 11, 17 and 18).

  1. The authors only presented the cell survival index data for the liposomal formulation of Ru (HoThyRu/DOTAP). It is recommended that the authors include information regarding the cell survival index when cells are treated with the Ru complex alone (AziRu).

We thank the reviewer for this remark highlighting a very interesting concern on the differences between the naked low molecular weight ruthenium(III) complex AziRu and the Ru-based final nanoformulation. This outcome has been discussed in our previous reports, where in vitro evaluations on both the nanostructured AziRu and naked AziRu were extensively performed. Indeed, it is an aspect that we have repeatedly underscored in our former works to emphasize the benefits, both in terms of safety and efficacy, of the use of the nanostructured AziRu with respect to the bare metal complex. For examples, please see ref. 11 (Irace et al.,  Sci Rep. 2017;7:45236. doi: 10.1038/srep45236) and 17 (Piccolo et al., Sci Rep. 2019;9(1):7006. doi: 10.1038/s41598-019-43411-3). Moreover, cellular treatments with the AziRu complex represents an investigational internal control that we perform routinely to check the progress of the experiments. Therefore, in line with the reviewer’s suggestion, and looking to fill in the data in the current paper, we have now included the results obtained by in vitro bioscreens on MDA-MB-231 cells treated with AziRu. Consequently, in the revised version of the manuscript we have improved Figure 1 (a and b) and the relative legend, as well as the experimental section (Bioscreens in vitro, line 160-161) and the results section (Antiproliferative effect in vitro, line 413-414).

  1. In addition to analyzing the mRNA levels of E-cadherin, N-cadherin, vimentin, Slug, and Snail, it is recommended that the authors also investigate the protein expression after treatment with HoThyRu/DOTAP.

We basically agree with the reviewer's recommendation. In fact, as evidenced by our previous research papers, we usually perform protein expression profiling by means of immunoblotting assays (WB analysis). However, the aim of the reported RT-qPCR investigations was just to support scientific evidence – illustrating the effect of our nanoformulation on cell migration and invasion - by means of preliminary data arising from expression studies of genes potentially involved in these processes. Indeed, the use of gene expression analysis tools has the potential of unravel how specific genes respond to drugs treatment, allowing to assume effects and responses on cellular dynamic process, as well as to clarify drug’s mechanisms of action. In this frame, while substantially agreeing with the reviewer, many researchers believe RT-qPCR analysis appropriate to provide indications concerning proteins gene expression even without exploring the final protein expression profile (for instance, see Freitas et al., Evaluation of reference genes for gene expression analysis by real-time quantitative PCR (qPCR) in three stingless bee species (Hymenoptera: Apidae: Meliponini). Sci Rep 9, 17692 (2019). https://doi.org/10.1038/s41598-019-53544-0). Moreover, the preliminary findings herein reported about EMT will pave the way for a forthcoming experimental study entirely focusing on the antimetastatic properties of the HoThyRu/DOTAP nanosystem and based exclusively on a proteomic approach. This is another reason why we have here opted to use a different experimental approach.

  1. Does the accumulation of nanoformulation in the kidney and liver cause any toxicities in these tissues?

We thank to the reviewer for allowing us to further delve into interesting topics concerning HoThyRu/DOTAP biocompatibility. We have already regarded these aspects in the discussion section of the current manuscript, highlighting the safety features of our experimental protocol (lane xx), which have been the subject of an in-depth study formerly conducted in the same animal model (female athymic nude Foxn1nu) and therefore not repeated here. In fact, in ref. 18 (Piccolo et al., Cancers. 2021;13(20):5164. doi: 10.3390/cancers13205164), the HoThyRu/DOTAP safety profile in vivo was extensively investigated. Haematological investigations including complete blood count (CBC) test and leukocyte formula, as well as liver and kidney toxicity test were performed. Blood diagnostic profile of HoThyRu/DOTAP treated animals revealed a physiological response both in terms of blood counts and biomarkers’ activities. In addition, autopsy tests documented by photographic evidence did not show acute toxicity in the main organs, including kidney and liver. Overall, no significant biochemical as well as clinically relevant alterations were observed in the treated animal group. As well for this study, after in vivo treatments we repeated the autopsy examination on the main internal organs of the animals, which we deliberately did not show since the data are very similar to those of ref. 18.

  1. typo is found: Title “Ruthe-nium”

        Thank you. Done. 

Round 2

Reviewer 2 Report

This reviewer has no further questions.